# Effect of Caffeinated Chewing Gum on Maximal Strength, Muscular Power, and Muscle Recruitment During Bench Press and Back Squat Exercises

**DOI:** 10.3390/nu17152455

**Published:** 2025-07-28

**Authors:** Li Ding, Jue Liu, Yixuan Ma, Tze-Huan Lei, Mathew Barnes, Li Guo, Bin Chen, Yinhang Cao, Olivier Girard

**Affiliations:** 1Key Laboratory of Exercise and Health Sciences (Ministry of Education), Shanghai University of Sport, Shanghai 200438, China; 2321111021@sus.edu.cn; 2School of Athletic Performance, Shanghai University of Sport, Shanghai 200438, China; xuanx121@163.com; 3Department of Rehabilitation Medicine, Huashan Hospital, Fudan University, Shanghai 200040, China; susanjue@126.com; 4College of Physical Education, Hubei Normal University, Huangshi 435002, China; tzehuanlei@gmail.com; 5School of Sport, Exercise and Nutrition, Massey University, Palmerston North 4443, New Zealand; m.barnes@massey.ac.nz; 6School of Exercise and Health, Shanghai University of Sport, Shanghai 200438, China; guoli@sus.edu.cn; 7Department of Public Physical Education, Fujian Agriculture and Forestry University, Fuzhou 350000, China; chenbin@fafu.edu.cn; 8School of Human Sciences (Exercise and Sport Science), The University of Western Australia, Perth 6800, Australia; oliv.girard@gmail.com

**Keywords:** ergogenic aid, nutritional supplement, neuromuscular function, one-maximum repetition, resistance exercise

## Abstract

**Background/Objectives**: This study aims to investigate the effects of caffeinated chewing gum on maximal strength, muscular power, and neural drive to the prime movers during bench press and back squat in resistance-trained men. **Methods**: Sixteen resistance-trained males participated in a double-blind, randomized trial, chewing either caffeinated gum (4 mg/kg) or placebo gum on two separate occasions, seven days apart. After chewing for 5 min, participants performed a maximal strength test followed by muscular power assessments at 25%, 50%, 75%, and 90% of their one-repetition maximum (1RM), completing with 3, 2, 1, and 1 repetition (s), respectively, for bench press and back squat. Surface electromyography data were recorded for each repetition. **Results**: Caffeinated gum did not significantly improve one-repetition maximum (1RM) for bench press (*p* > 0.05), but increased mean frequency (MF) and median frequency (MDF) in anterior deltoid, pectoralis major, and biceps brachii (all *p* < 0.05) compared to placebo. For back squat, 1RM increased with caffeinated gum, along with higher MF and MDF in vastus medialis (all *p* < 0.05). Caffeinated gum also improved mean and peak velocities, and mean and peak power outputs at 25–75% 1RM during the bench press (all *p* < 0.05), along with elevated MDF in pectoralis major and biceps brachii (all *p* < 0.05). Similar improvements were seen in mean and peak velocities during the back squat at 25–90% 1RM (all *p* < 0.05), along with higher MF and MDF in vastus medialis and increased normalized root mean square activity in gluteus maximus (all *p* < 0.05). **Conclusions**: Caffeinated chewing gum (4 mg/kg) enhanced muscular power (25–75% 1RM) in the bench press and improved maximal strength and muscular power (25–90% 1RM) in the back squat by increasing muscle recruitment in resistance-trained men.

## 1. Introduction

The bench press and back squat are commonly prescribed in strength and conditioning programs to develop upper and lower body muscle strength and power [1,2]. In recent years, advanced training methods such as velocity-based training, repetitions in reserve, and blood flow restriction have been developed to improve their execution [3,4]. Meanwhile, nutritional ergogenic aids have gained significant attention as alternative strategies to boost performance [5,6,7].

Caffeine (1,3,7-trimethylxanthine), classified as a nutritional supplement by the International Olympic Committee [8], has been shown to enhance aerobic exercise [9], anaerobic exercise [10], and sport-specific performance [11,12] in both males and females. Recent evidence indicates that caffeine (3–6 mg/kg) significantly enhances maximal strength (one-repetition maximum [1RM]) [5,10,13] and muscular power (barbell velocity and/or power output) at low-to-high loads (25–75% 1RM) [14,15] in the bench press and back squat exercises for resistance-trained individuals, with more pronounced effects typically observed in the back squat [10,15]. Surface electromyography (sEMG) evidence, primarily from single-joint resistance exercises (e.g., knee extensions, elbow flexions), shows that caffeine enhances muscle strength by increasing muscle recruitment [16,17]. However, sEMG data from multi-joint exercises like the bench press and back squat, which involve coordination of multiple muscle groups, are limited, restricting understanding of the neuromuscular mechanisms behind caffeine’s effects. Furthermore, some studies suggest that larger lower-limb muscle groups (e.g., knee extensors) may experience greater improvements in motor unit recruitment than smaller upper-limb muscles (e.g., elbow flexors) [10,15,18]. Therefore, to explain the disparity in caffeine’s effects between the back squat and bench press, sEMG data from both exercises are needed to substantiate this claim.

Most studies administer caffeine in capsule form [5,13,14], which has drawbacks, including slow absorption (45–60 min) and potential gastrointestinal irritation during intense activity [19]. Caffeinated chewing gum represents a viable alternative, as caffeine is absorbed directly through the oral mucosa, providing a faster absorption rate (5–10 min) and reducing gastrointestinal irritation [20,21,22]. Mounting research demonstrates its performance-enhancing effect on endurance, repeated sprints, and countermovement jumps [23,24,25]. In recent years, increasing attention has been given to its effects on resistance exercise performance, with studies noting increases in either maximal strength [26] or muscular power [27,28] in resistance-trained men. Recent reviews suggest that caffeine may have a more pronounced effect on muscular power than maximal strength during resistance exercises [29,30]. However, no studies have measured both maximal strength and muscular power within the same trial, highlighting a significant research gap.

This study aims to investigate the effects of caffeinated chewing gum on maximal strength, muscular power, and neural drive to the prime movers during bench press and back squat in resistance-trained men. We hypothesize that caffeinated chewing gum will improve maximal strength and muscular power by increasing muscle recruitment in both exercises. Our research provides sEMG-based insights into the neuromuscular mechanisms underlying the effects of caffeine on resistance exercise and offers practical guidance to optimize caffeine supplementation in resistance training.

## 2. Materials and Methods

### 2.1. Participants

A priori power analysis, conducted using G*Power software (version 3.1; Universität Düsseldorf, Düsseldorf, Germany), indicated that 10 participants would be sufficient (alpha: 0.05, power: 0.80, correlation coefficient: 0.90 [26], and effect size: 0.18–1.28 for 1RM and muscular power in resistance exercise [14,26,31]). To account for potential dropouts, 16 resistance-trained men were recruited (mean ± SD: age 20.8 ± 1.6 yr, height 173.5 ± 5.1 cm, body mass 72.9 ± 6.1 kg, fat body mass 11.2 ± 3.0 kg, lean body mass 61.7 ± 5.3 kg, training history 4.2 ± 1.9 yr, 1RM bench press 92.0 ± 12.4 kg, 1RM back squat 152.5 ± 20.6 kg). Habitual caffeine intake (41.4 ± 46.5 mg/day) was assessed using a validated self-reported questionnaire [32], and participants were classified as either naive or mild caffeine consumers (0–2.99 mg/kg/day) [33].

Participants were required to meet the following criteria: (1) healthy men aged 18–35 years; (2) at least two years of resistance training experience, with a minimum of three sessions per week during the three months preceding the experiment; and (3) ability to execute the bench press and back squat with loads of at least 100% and 125% of body mass, respectively [10]. Exclusion criteria included: (a) any neuromuscular, immunological, cardiometabolic, neurological, or musculoskeletal disorders; (b) self-reported smoking; or (c) caffeine allergy. Each participant signed a written informed consent form. The experimental procedures were approved by the Scientific Research Ethics Committee of Shanghai University of Sport (No. 102772023RT203) on 15 March 2024.

### 2.2. Study Design

This double-blind, placebo-controlled, crossover, randomized study involved three laboratory visits. The first visit included familiarization and 1RM assessments for bench press and back squat. The subsequent visits were for the experimental sessions, where participants randomly received either caffeinated gum (4 mg/kg) or placebo gum before completing the same assessments: maximal strength (1RM) and muscular power at 25%, 50%, 75%, and 90% 1RM for both exercises. Randomization and counterbalancing of the intervention order were performed using randomization software (Excel Office, Microsoft, WA, USA) [34]. Visits were separated by 3–7 days to ensure full recovery and scheduled between 12:00 and 16:00 to minimize circadian effects [13]. Participants were required to avoid intense exercise and caffeine (e.g., coffee, chocolate, soda, and energy drinks) 24 h prior to each visit.

### 2.3. Experimental Protocol

During the familiarization session, participants’ height, body weight, and body composition were measured using electric bioimpedance (X-scan Plus II, Busan, Korea). One experienced personal trainer then assessed each participant’s 1RM for the bench press and back squat using free-weight equipment (Cybex, Medway, MA, USA), following the guidelines of Baechle and Earle [35]. The 1RM values were used to determine individualized loads corresponding to 25%, 50%, 75%, and 90% 1RM for both exercises. Participants’ dietary and physical activity habits from the previous 24 h were recorded using a 24-h recall questionnaire and the International Physical Activity Questionnaire (IPAQ) [36], and these habits were maintained consistently throughout the subsequent sessions.

For the experimental session, upon arrival at the laboratory, participants performed a standardized 10-min warm-up, which comprised 5 min of self-selected cycling on an ergometer followed by 5 min of joint mobilization and dynamic stretching exercises. Subsequently, participants chewed either caffeinated or placebo gum for 5 min while sEMG electrodes were affixed. The testing protocol (Figure 1) began with measuring 1RM, followed by muscular power measurements at incremental loads (25%, 50%, 75%, and 90% 1RM) during bench press and back squat exercises. The order of the exercises was counterbalanced across participants and remained consistent for each individual during both sessions, using randomization software (Excel, Microsoft, WA, USA). Two experienced personal trainers ensured proper technique adherence during the exercises [35]. Participants were prohibited from wearing weightlifting belts, bench shirts, or other supportive garments during testing.

### 2.4. Maximal Strength (1RM) Test

Participants first performed a warm-up consisting of 10 and 5 repetitions at 50% and 75% 1RM, respectively, as assessed during the familiarization session [5]. Following the recommendations of Baechle and Earle [35], 1RM was determined within 3–5 attempts, with a 5-min rest between successful attempts. Ratings of perceived exertion (RPE) and pain perception were measured by the 6–20 Borg scale and a 0–10 numeric pain rating scale [37,38], respectively, within 5 s of each successful attempt. Pilot testing showed *excellent* interclass correlation coefficients (ICC) for 1RM measurements for the bench press (ICC = 0.98, 95% CI = 0.96–0.99) and back squat (ICC = 0.98, 95% CI = 0.95–0.99) across three days.

### 2.5. Muscular Power Test

After a 5-min rest following 1RM determination, participants performed the same exercises at 25%, 50%, 75%, and 90% of their 1RM, measured in familiarization session, using a 2/0/X/0 cadence (2 s for the eccentric phase, no pause during the transition, X representing maximum concentric tempo, and 0 s at the end of the movement) [39,40], stabilized by a metronome set to 30 beats/min. Participants completed three repetitions at 25% 1RM, two repetitions at 50% 1RM, and one repetition at 75% and 90% 1RM, with 3-min rest intervals. To assess the impact of caffeine on muscular power at different loads, two additional sets were performed at 25% and 50% of the post-ingestion 1RM, with three and two repetitions, respectively. The GymAware Power Testing system (Kinetic Performance Technologies, Canberra, Australia), which has demonstrated *excellent* reliability in measuring velocity and power output [41], was used to record bar displacement during the concentric phase, including mean velocity (MV in m/s), peak velocity (PV in m/s), mean power output (MPO in W), and peak power output (PPO in W). RPE and pain perception were recorded within 5 s after the final repetition. Pilot testing indicated *excellent* interclass ICC for MV (bench press: ICC = 0.89, 95%CI = 0.82–0.93; back squat: ICC = 0.89, 95% CI = 0.82–0.94) and MPO (bench press: ICC = 0.96, 95% CI = 0.94–0.98; back squat: ICC = 0.94, 95% CI = 0.91–0.97) across three days of testing.

### 2.6. Surface Electromyography (sEMG)

sEMG data were recorded using a surface EMG system (Noraxon, Inc., Scottsdale, AZ, USA) at a 2,000 Hz sampling rate. To ensure consistency across sessions, electrode placement was marked with an anatomical pen mark. Skin impedance was minimized by shaving, abrading with sandpaper, and cleaning with alcohol pads. Following SENIAM project’s guidelines [42], self-adhesive electrodes (sEMG Electrodes AE-131, NeuroDyne Medical, MA, USA) were placed on the muscle bellies of the dominant side’s *pectoralis major*, *anterior deltoid*, *posterior deltoid*, *biceps brachii*, lateral head of the *triceps* for the bench press, and on the *gluteus maximus*, *rectus femoris*, *vastus medialis*, *vastus lateralis*, *biceps femoris*, and *tibialis anterior* for the back squat. Electrodes were secured using adhesive tape. An electronic goniometer (Biometrics Ltd., Gwent, UK) was used to monitor knee and elbow joint flexion and extension, characterizing the concentric and eccentric phases of each repetition.

All raw sEMG signals were analyzed using MATLAB (Version R2021a; MathWorks Inc., Natick, MA, USA). sEMG data corresponding to the concentric phase of each repetition were extracted using electronic goniometer data [43] and filtered with a zero-phase, fourth-order Butterworth bandpass filter (10–500 Hz) [44]. A sliding window (0.100 s length, 0.08 s overlap) was employed to calculate RMS values, reflecting motor unit recruitment [45]. For the maximal strength test, RMS values were not normalized due to fixed electrode placement and within-subject repeated-measures comparisons [46]. For the muscular power test, RMS values at different loads were normalized to the peak RMS values from the post-substance ingestion 1 RM test [44]. Normalized RMS values for each load (25%, 50%, 75% and 90% 1 RM) were averaged across the respective repetitions [44]. Additionally, fast Fourier transformation was applied to determine mean frequency (MF) and median frequency (MDF) for both maximal strength and muscular power tests [47], which reflect muscle recruitment strategy (proportion of type I or II muscle fiber recruitment) [48].

### 2.7. Supplementation Protocol

Participants chewed gum for 5 min, with the duration measured using a stopwatch, before discarding it in a designated container. Previous research has confirmed that this duration releases approximately 85% of the caffeine contained in the gum [21]. The caffeine source selected was commercially available Military Energy Gum (Market Right Inc., Plano, IL, USA), containing 100 mg of caffeine per piece. The placebo gum was a caffeine-free version with a similar taste and shape. To achieve a relative dose of 4 mg/kg, which has been shown to enhance performance with minimal side effects [9], all gum samples were crushed, ground, blended, and reshaped. These were then weighed precisely using a high-precision electronic digital scale and wrapped in aluminum foil by a designated experimenter.

### 2.8. Blinding and Side Effects Assessment Protocol

To evaluate blinding effectiveness, participants were asked, “*What substance do you think you ingested*?” immediately after chewing gum and again post-exercise. Response options included: (a) caffeinated chewing gum; (b) placebo chewing gum, or (c) I don’t know [49]. Additionally, participants completed the Side Effects Questionnaire, a nine-item dichotomous scale to evaluate potential caffeine side effects [50], immediately and 24 h after the exercise sessions.

### 2.9. Statistical Analyses

Data were analyzed using SPSS software (Version 22.0; SPSS Inc., Chicago, IL, USA). The Shapiro–Wilk test confirmed normal distribution of all variables. Paired sample t-tests were used to compare exercise performance (1RM), sEMG signals (RMS, MF, and MDF), and perceptual responses (RPE and pain perception) between caffeine and placebo conditions during the maximal strength test. Additionally, a two-way repeated measure ANOVA (load [25%, 50%, 75% and 90% 1RM] × condition [caffeine and placebo]) was performed to analyze the effects of caffeinated chewing gum on exercise performance (MV, PV, MPO, and PPO), sEMG signals (RMS, MF, and MDF), and perceptual responses (RPE and pain perception). Partial eta squared (*ηp*^2^) values were calculated to estimate effect sizes for main effects. Significant main effects were followed by *Bonferroni*-corrected post hoc tests. Cohen’s *d* was calculated for pairwise comparisons to assess effect sizes, categorized as *trivial* (<0.20), *small* (0.20–0.49), *moderate* (0.50–0.79), and *large* (≥0.80) [51]. Additionally, the Bang’s Blinding Index (BBI) was used to assess blinding effectiveness, and McNemar’s test was applied to detect variations in side effects between conditions. Results are presented as mean ± standard deviation (SD), with statistical significance set at *p*  < 0.05.

## 3. Results

### 3.1. Maximal Strength and sEMG Assessment

For the bench press, caffeinated chewing gum did not significantly increase 1RM (+2.1 ± 5.0%, *p* = 0.118, *d =* 0.11) (Figure 2). However, it increased MF in the *pectoralis major* (+8.8 ± 13.1%, *p* < 0.05, *d =* 0.55), *anterior deltoid* (+7.8 ± 10.0%, *p* < 0.05, *d =* 0.49), *posterior deltoid* (+5.5 ± 14.3%, *p* = 0.050, *d =* 0.57), and *biceps brachii* (+8.3 ± 15.7%, *p* < 0.05, *d =* 0.46) and MDF in the *anterior deltoid* (+9.7 ± 11.7%, *p* < 0.05, *d =* 0.74) compared to placebo (Table 1). During the bench press, RMS did not differ between conditions for any upper-body muscles (all *p* > 0.05) (Table 1).

For the back squat, caffeinated chewing gum significantly increased 1RM (+5.0 ± 6.0%, *p* < 0.05, *d =* 0.39) compared to placebo (Figure 2), with higher MF (+7.7 ± 10.4%, *p* < 0.05, *d =* 0.59) and MDF (+9.0 ± 13.0%, *p* < 0.05, *d =* 0.49) in the *vastus medialis* (Table 1). In addition, RMS did not differ between conditions for any lower body muscles during back squat (all *p* > 0.05) (Table 1).

### 3.2. Muscular Power and sEMG Assessment

For the bench press, a significant main effect of condition was observed for MV (*F* = 33.252, *p* < 0.05, *ηp*^2^ = 0.69), PV (*F* = 15.924, *p* < 0.05, *ηp*^2^ = 0.52), MPO (*F* = 23.988, *p* < 0.05, *ηp*^2^ = 0.615), and PPO (*F* = 19.474, *p* < 0.05, *ηp*^2^ = 0.57) (Figure 3 and Figure 4). Post-hoc comparisons revealed that caffeinated chewing gum significantly increased MV, PV, MPO, and PPO at 25% 1RM (MV: +8.7 ± 9.6%, *p* < 0.05, *d* = 0.68; PV: +7.0 ± 9.2%, *p* < 0.05, *d* = 0.49; MPO: +14.4 ± 15.4%, *p* < 0.05, *d* = 0.68; PPO: +11.2 ± 14.8%, *p* < 0.05, *d* = 0.71), 50% 1RM (MV: +9.0 ± 10.4%, *p* < 0.05, *d* = 0.62; PV: +7.5 ± 10.0%, *p* < 0.05, *d* = 0.46; MPO: +10.2 ± 11.6%, *p* < *0.05*, *d* = 0.74; PPO: +7.5 ± 13.7%, *p* < 0.05, *d* = 0.51), and 75% 1RM (MV: +7.0 ± 11.8%, *p* < 0.05, *d* = 0.25; PV: +9.8 ± 13.5%, *p* < 0.05, *d* = 0.35; MPO: +6.6 ± 11.7%, *p* < 0.05, *d* = 0.33; PPO: +11.2 ± 18.3%, *p* < 0.05, *d* = 0.52), but not at 90% 1RM (MV: +10.5 ± 35.9%, *p* = 0.843, *d* = 0.03; PV: +0.5 ± 6.1%, *p* = 0.274, *d* = 0.26; MPO: +11.3 ± 38.9%, *p* = 0.763, *d* = 0.06; PPO: +10.0 ± 33.2%, *p* = 0.769, *d* = 0.10) compared to placebo (Figure 3 and Figure 4).

Regarding the sEMG analysis, a significant main effect of condition was observed for MF in the *biceps brachii* (*F* = 9.418, *p* < *0.05*, *ηp*^2^ = 0.39), as well as for MDF in the *pectoralis major* (*F* = 7.482, *p* < 0.05, *ηp*^2^ = 0.33) and *biceps brachii* (F = 6.088, *p* < 0.05, *ηp*^2^ = 0.29) (Table 2. Post-hoc comparisons showed that caffeinated chewing gum increased MF for the *biceps brachii* at 25% 1RM (+6.5 ± 13.8%, *p* < 0.05, *d* = 0.64) and 50% 1RM (+6.1 ± 15.5%, *p* < 0.05, *d* = 0.61). Additionally, higher MDF were noted for the *pectoralis major* at 50% 1RM (+11.1 ± 16.4%, *p* < 0.05, *d* = 0.87), 75% 1RM (+13.1 ± 19.2%, *p* < 0.05, *d* = 0.82), and 90% 1RM (+9.3 ± 12.1%, *p* < 0.05, *d* = 0.80) after consuming caffeinated chewing gum (Table 2). Likewise, higher MDF was observed for the *biceps brachii* at 25% 1RM (+9.6 ± 23.0%, *p* < 0.05, *d* = 0.67), 50% 1RM (+7.2 ± 18.3%, *p* < 0.05, *d* = 0.54), and 75% 1RM (+4.3 ± 23.4%, *p* < 0.05, *p* < 0.05, *d* = 0.17) compared to placebo (Table 2). However, normalized RMS did not differ between conditions for any upper-body muscles during bench press (all *p* > 0.05) (Table 2).

For the back squat, a significant main effect of condition was observed for MV (*F* = 15.767, *p* < *0.05*, *ηp*^2^ = 0.51), PV (*F* = 14.693, *p* < *0.05*, *ηp*^2^ = 0.50), MPO (*F* = 19.378, *p* < *0.05*, *ηp*^2^ = 0.56), and PPO (*F* = 11.327, *p* < *0.05*, *ηp*^2^ = 0.43) (Figure 3 and Figure 4). Post-hoc comparisons showed that caffeinated chewing gum significantly increased MV, PV, and MPO at 25% 1RM (MV: +6.7 ± 8.9%, *p* < 0.05, *d* = 0.72; PV: +8.4 ± 12.0%, *p* < 0.05, *d* = 0.66; MPO: +8.2 ± 10.7%, *p* < 0.05, *d* = 0.50), 50% 1RM (MV: +8.8 ± 10.4%, *p* < *0.05*, *d* = 0.77; PV: +7.8 ± 10.6%, *p* < 0.05, *d* = 0.62; MPO: +8.8 ± 10.8%, *p* < 0.05, *d* = 0.47), 75% 1RM (MV: +12.3 ± 13.2%, *p* < *0.05*, *d* = 0.66; PV: +6.7 ± 9.7%, *p* < 0.05, *d* = 0.54; MPO: +12.4 ± 12.9%, *p* < *0.05*, *d* = 0.52), and 90% 1RM (MV: +20.8 ± 20.6%, *p* < *0.05*, *d* = 0.86; PV: +18.2 ± 31.8%, *p* < 0.05, *d* = 0.81; MPO: +20.0 ± 20.9%, *p* < *0.05*, *d* = 0.72) compared to placebo (Figure 3 and Figure 4). Additionally, caffeine chewing gum improved PPO at 75% 1RM (+10.1 ± 15.1%, *p* < 0.05, *d* = 0.49) and 90% 1RM (+13.3 ± 13.2%, *p* < *0.05*, *d* = 0.73).

Regarding the sEMG analysis, a significant main effect of condition was observed for MF (*F* = 5.503, *p* < 0.05, *ηp*^2^ = 0.27) and MDF (*F* = 4.851, *p* < 0.05, *ηp*^2^ = 0.24) for the *vastus medialis* (Table 2). Post-hoc comparisons showed that caffeinated chewing gum increased MF for the *vastus medialis* at 25% 1RM (+9.7 ± 11.2%, *p* < *0.05*, *d =* 0.92), 50% 1RM (+3.7 ± 9.3%, *p* < 0.05, *d =* 0.41), and 75% 1RM (+6.8 ± 12.0%, *p* < 0.05, *d =* 0.60), as well as MDF at 25% 1RM (+11.7 ± 12.1%, *p* < *0.05*, *d =* 0.79) compared to placebo gum (Table 2). Furthermore, a significant main effect of condition was found for normalized RMS in the *gluteus maximus* (*F* = 7.102, *p* < 0.05, *ηp*^2^ = 0.321) (Table 2). Post-hoc comparisons revealed that caffeinated chewing gum significantly increased normalized RMS for the *gluteus maximus* at 25% 1RM (+23.0 ± 40.1%, *p* < 0.05, *d* = 0.66), 50% 1RM (+21.4 ± 42.0%, *p* < 0.05, *d* = 0.76), and 75% 1RM (+23.1 ± 33.6%, *p* < 0.05, *d* = 0.86), compared to placebo (Table 2).

### 3.3. Perceptual Responses

RPE and pain perception did not differ between conditions during either the maximal strength or muscular power tests for both the bench press and back squat exercises (all *p* > 0.05) (Table 3 and Table 4).

### 3.4. Assessment of Blinding and Side Effects

Blinding was effective immediately following ingestion (caffeine BBI: 0.19 [95%CI: −0.14–0.52]; placebo BBI: −0.13 [95% CI: −0.37–0.12]) and post-exercise (caffeine BBI: 0.27 [95%CI: −0.13–0.66]; placebo BBI: 0.13 [95% CI: −0.20–0.45]). Caffeinated chewing gum significantly increased *perception of performance improvement* and feeling of *increased vigor/activeness* compared to placebo (all *p* < *0.05*), but did not increase other side effects (e.g., *headache, muscle soreness*, and *anxiety*) (all *p* > 0.05) (Table 5). At 24-h post-exercise, it *increased urine output* (*p* < 0.05), with no significant effects on other side effects (e.g., *headache*, *muscle soreness, and insomnia*) (all *p* > 0.05) (Table 5).

## 4. Discussion

The major findings indicate that caffeinated chewing gum (4 mg/kg) significantly improved muscular power in the bench press and both maximal strength and muscular power in the back squat. This study is the first to demonstrate that caffeinated chewing gum also increases muscle recruitment in the active musculature during both exercises. These results support our hypothesis, demonstrating that caffeinated chewing gum enhances performance by augmenting neural drive to the prime movers in both bench press and back squat exercises.

### 4.1. Effects of Caffeinated Chewing Gum on Maximal Strength

Caffeinated chewing gum enhanced 1RM for back squat (Figure 2) and increased MF and MDF of *vastus medialis* (Table 1). It also increased MF and MDF in the *anterior deltoid*, *pectoralis major*, and *biceps brachii* (all *p* < 0.05) during bench press. These increases in MF and MDF indicate greater recruitment of fast-twitch (type II) muscle fibers, likely due to increased cortico-spinal tract excitability [48]. Additionally, caffeine may reduce fatigue and pain perceptions, contributing to improved strength [5,29]. Our findings support this, as caffeine enhanced 1RM for back squat without affecting RPE or pain perception. Moreover, the bitter taste of caffeinated chewing gum could activate brain regions involved in motor control, thereby enhancing central arousal and muscle strength [52,53]. Caffeine may also improve muscle contractility by enhancing sodium-potassium and calcium pump function [21,54], though this warrants verification through percutaneous nerve stimulation.

Our results revealed that caffeinated chewing gum improved 1RM for back squat in resistance-trained men (Figure 2). However, a previous study has reported conflicting results, where caffeine supplementation (400 mg) in tablet form did not enhance 1RM for leg extension in untrained participants [55], possibly due to differences in training status. Trained individuals likely have a higher density of adenosine receptors, making them more responsive to caffeine and potentially benefiting more from its effects on muscle strength [56]. Additionally, the mixed supplement used by Hendrix et al. [55] contained ten other substances, including cinnamon, black tea extract, and ginger extract, complicating the isolation of caffeine effects. Furthermore, our study found that caffeinated chewing gum did not improve 1RM for the bench press (Figure 2). This may be attributed to caffeine-induced increases in antagonist muscle activity, as demonstrated by higher MF in the *posterior deltoid* and *biceps brachii*, which could have counteracted the performance-enhancing effects.

### 4.2. Effects of Caffeinated Chewing Gum on Muscular Power

Caffeinated chewing gum enhanced MPO, PPO, MV, and PV at 25–75% 1RM in the bench press, and improved MPO, MV, and PV at 25–90% 1RM, as well as PPO at 75–90% 1RM in the back squat (Figure 3 and Figure 4). Additionally, it increased MDF and MF in the *vastus medialis*, *pectoralis major*, and *biceps brachii* (Table 2). Building on previous research showing the impact of caffeinated chewing gum (300 mg) on upper-limb muscular power during bench press [27], our results confirm its positive effect on lower-limb muscular power during back squat. The increase in MDF and MF suggests that caffeine may enhance muscular power by improving motor unit recruitment, similar to its effects on 1RM performance. Specifically, caffeine could increase central nervous system excitability through antagonistic actions [57], thereby recruiting more fast-twitch muscle fibers and improving movement velocity. Additionally, caffeine elevated RMS activity of the *gluteus maximus* during back squat (Table 2), suggesting greater motor unit recruitment and/or higher firing frequency [47,58]. Overall, these results demonstrate that caffeine boosts muscular power by optimizing muscle recruitment strategies, further elucidating the neuromuscular mechanisms underlying its effects in the bench press and back squat.

Our study showed that 4 mg/kg of caffeine in gum significantly improved MV at 25–90% 1RM during back squat (Figure 3). In contrast, a previous study found that caffeine (3 mg/kg) in capsule form significantly increased MV at 25–75% 1RM during back squat, with higher doses (6–9 mg/kg) required to enhance MV at 90% 1RM (10). This suggests that caffeinated chewing gum may enhance muscular power at higher loads (90% 1RM) with a lower dose (4 vs. 6–9 mg/kg) during back squat compared to a capsule [14]. Additionally, our results showed that caffeinated chewing gum significantly increased PV at 25–75% 1RM in the bench press and 25–90% 1RM in the back squat in resistance-trained men (Figure 3). This contrasts with Ruiz-Fernández et al. [15], who found no PV improvements across similar intensities. This discrepancy may be attributed to the inclusion of female participants (10 out of 20) in their study, while we exclusively recruited males. Hormonal fluctuations during the menstrual cycle may impair caffeine metabolism, potentially reducing its ergogenic effects in women [59].

Recent reviews have suggested that caffeine may have larger effects on muscular power than on maximal strength (1RM) during resistance exercise [29,30]. Our results support this, showing that the effect sizes (*d*) for both mean and peak velocity were larger than for 1RM in both the bench press (0.25–0.62 *vs*. 0.11) and back squat (0.54–0.86 *vs*. 0.39) exercises. This may be because the load during muscular power tests is lower than during maximal strength tests, allowing for greater potential improvements in motor unit recruitment. Further supporting this, caffeine enhanced the RMS activity of the *gluteus maximus* at 25–75% of 1RM, but did not increase the RMS of any muscle at 90% 1RM or during maximal strength tests. Additionally, previous studies on the effects of caffeine on muscular power during bench press and back squat at varying loads (25–90% 1RM) typically used 1RM values measured before caffeine ingestion (i.e., pre-experimental session) [14,15]. However, both our study and prior research [60] show that caffeine can significantly increase 1RM, which may lower the relative load during muscular power tests. Therefore, it is important to explore whether the effects of caffeine on muscular power at different loads are mediated by its impact on 1RM. Our results (Figure A1) show that caffeine significantly enhances MV and PV at 25–50% 1RM measured both pre- and post-ingestion. Overall, this suggests that the performance-enhancing effects of caffeine on muscular power during bench press and back squat are not mediated by improvements in 1RM.

### 4.3. Difference in Performance Improvements Between Bench Press and Back Squat

Caffeinated chewing gum significantly improved 1RM and both MV and PV at 90% 1RM during back squat, but had no effects on these metrics during bench press at the same load. Consequently, caffeinated chewing gum has a larger effect on muscular strength and power in the back squat compared to the bench press. Similarly, Grgic et al. [10] reported that ingesting caffeine (6 mg/kg) in capsule form increased 1RM for the back squat but had no effect on the bench press in resistance-trained men. The differential effects between the upper and lower limbs may be attributed to differences in muscle mass [15], with larger muscle groups (i.e., knee extensors) offering more potential for improved activation than smaller muscle groups (i.e., elbow flexors) [61]. Our sEMG data support this, showing that caffeinated chewing gum increased RMS activity in the *gluteus maximus* during back squat, but did not affect upper-body muscles during the muscular power test.

A comparison of caffeine’s effects on bench press and back squat performance could help clarify its mechanism of action (central vs. peripheral) [15]. If caffeine primarily influences the central nervous system, its effects would likely be more pronounced during the back squat due to the greater muscle mass involved, which requires greater motor unit recruitment. Conversely, if caffeine acts through peripheral mechanisms, its effects would likely be comparable between exercises [62,63]. Our findings likely support the central effect hypothesis, as caffeine demonstrated a greater ergogenic effect on the back squat than on the bench press, suggesting that caffeine enhances force and power production by augmenting neural drive to active musculature.

### 4.4. Side Effects of Caffeinated Chewing Gum

The risk-benefit ratio of acute caffeine intake has been largely debated. Our results indicate that caffeinated chewing gum significantly enhanced the *perception of performance improvement* and the *increased vigor and activeness*, compared to placebo (Table 5). However, it did not have common side effects such as *headaches*, *palpitations,* and *gastrointestinal discomfort* (Table 5). While a recent review suggested that acute caffeine supplementation may negatively affect post-exercise recovery [64], our results show that, 24 h after exercise, caffeine only increased urine output, with no change in other side effects (i.e., insomnia, anxiety, and muscle soreness) (Table 5). Overall, caffeinated chewing gum may offer a relevant solution for athletes needing to compete multiple times within a short recovery period.

### 4.5. Limitations

This study has several limitations. First, the results are limited to resistance-trained males, reducing generalizability to other populations such as adolescents, older adults, and females. Second, sEMG provides a global estimate of neuromuscular activation but cannot distinguish between the number of recruited motor units or their firing frequencies, and crosstalk from neighboring muscles is possible [65]. Third, our study did not account for genetic variability in caffeine response. Previous research has found that individuals with the AA genotype are more sensitive to caffeine than those with AC or CC genotypes [9]. Finally, while randomized condition orders, consistent trial times, and dietary instructions were used to minimize variability, the lack of strict control over food intake 2–3 h before trials may have introduced variability between conditions.

### 4.6. Practical Implications

Our findings demonstrate that chewing caffeinated gum (4 mg/kg) for 5 min prior to training significantly increases 1RM, muscular power, and muscle recruitment during resistance exercises. Furthermore, given its time efficiency and minimal side effects, caffeinated gum is recommended as an effective and safe supplementation strategy to optimize resistance training performance.

## 5. Conclusions

In resistance-trained men, caffeinated chewing gum (4 mg/kg) enhanced muscular power at 25–75% 1RM in the bench press and improved maximal strength and muscular power at 25–90% 1RM in the back squat, while also increasing muscle recruitment in prime movers. Practically, caffeinated chewing gum offers a time-efficient and effective supplement for improving resistance exercise performance, with minimal risk of side effects.

## Figures and Tables

**Figure 1 nutrients-17-02455-f001:**
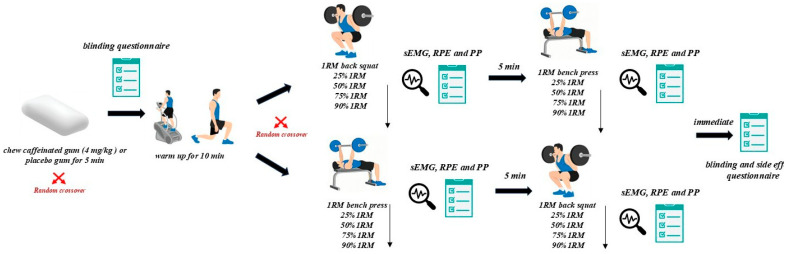
Experimental protocol overview.

**Figure 2 nutrients-17-02455-f002:**
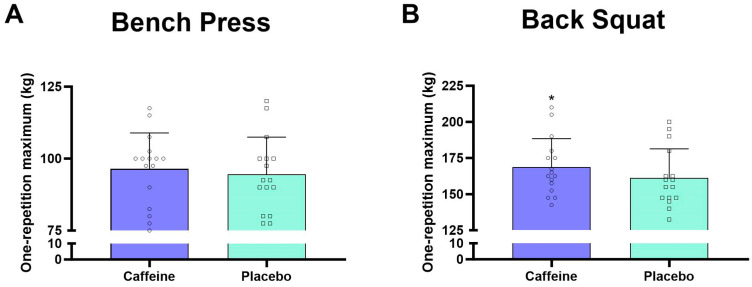
One-repetition maximum in the bench press (**A**) and back squat (**B**). Mean change in one-repetition maximum as a percentage for different exercise types. * *p* < 0.05 vs. placebo.

**Figure 3 nutrients-17-02455-f003:**
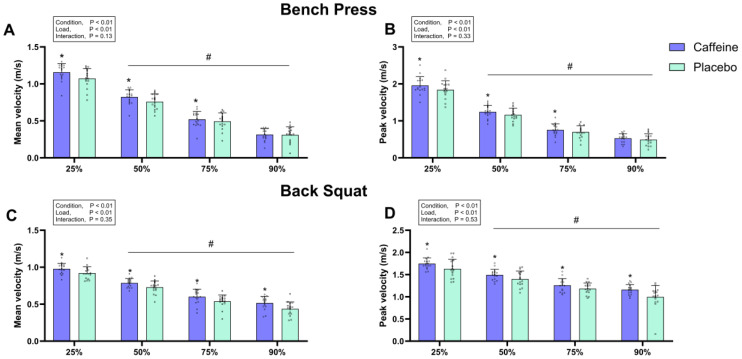
Mean and peak velocity in the bench press (**A**,**B**) and back squat (**C**,**D**). Error bars represent standard deviation. * *p* < 0.05 vs. placebo, # *p* < 0.05 vs. 25% 1RM.

**Figure 4 nutrients-17-02455-f004:**
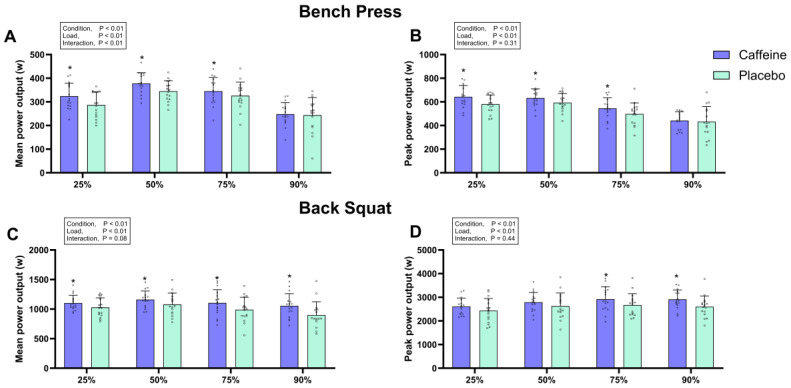
Mean and peak power output in the bench press (**A**,**B**) and back squat (**C**,**D**). Error bars represent standard deviation. * *p* < 0.05 vs. placebo.

**Table 1 nutrients-17-02455-t001:** Differences in mean frequency (MF), median frequency (MDF), and root mean square (RMS) between caffeine and placebo conditions during maximal strength tests in the bench press and back squat.

Indicators		Caffeine	Placebo
MF (Hz)	**bench press**		
	pectoralis major	114.77 ± 8.36 *	109.14 ± 11.95
	anterior deltoid	103.85 ± 11.97 *	97.77 ± 12.84
	posterior deltoid	135.81 ± 11.64 *	127.31 ± 16.81
	biceps brachii	103.89 ± 14.45 *	97.04 ± 15.46
	lateral head of the triceps	115.52 ± 10.05	115.33 ± 18.75
	**back squat**		
	gluteus maximus	96.06 ± 12.92	96.30 ± 14.84
	rectus femoris	138.73 ± 7.33	138.43 ± 13.24
	vastus medialis	135.81 ± 11.66 *	127.31 ± 16.81
	vastus lateralis	117.20 ± 19.52	110.05 ± 12.14
	biceps femoris	135.86 ± 18.36	135.20 ± 13.27
	tibialis anterior	175.74 ± 19.44	179.03 ± 14.09
MDF (Hz)	**bench press**		
	pectoralis major	87.12 ± 9.67 *	80.43 ± 11.36
	anterior deltoid	80.36 ± 9.71 *	73.58 ± 8.54
	posterior deltoid	71.83 ± 6.91	73.14 ± 13.03
	biceps brachii	73.46 ± 13.15	68.94 ± 13.24
	lateral head of the triceps	88.52 ± 8.25	88.56 ± 16.82
	**back squat**		
	gluteus maximus	65.49 ± 11.29	65.30 ± 12.12
	rectus femoris	108.09 ± 11.74	106.65 ± 15.79
	vastus medialis	107.16 ± 13.85 *	99.66 ± 16.86
	vastus lateralis	92.61 ± 14.52	86.32 ± 9.84
	biceps femoris	98.30 ± 23.35	99.32 ± 12.85
	tibialis anterior	153.97 ± 30.00	157.47 ± 21.52
RMS	**bench press**		
	pectoralis major	539.087 ± 243.322	507.364 ± 152.605
	anterior deltoid	1242.864 ± 532.495	1204.086 ± 422.809
	posterior deltoid	106.594 ± 24.952	196.141 ± 440.537
	biceps brachii	168.001 ± 60.284	195.402 ± 129.067
	lateral head of the triceps	1286.349 ± 699.629	1276.917 ± 538.298
	**back squat**		
	gluteus maximus	241.989 ± 91.331	258.463 ± 188.696
	rectus femoris	424.604 ± 159.942	401.519 ± 145.232
	vastus medialis	516.985 ± 183.246	519.793 ± 184.793
	vastus lateralis	737.438 ± 279.843	766.806 ± 225.589
	biceps femoris	246.338 ± 473.980	141.424 ± 67.653
	tibialis anterior	358.288 ± 222.216	293.237 ± 134.362

* *p* < 0.05 vs. placebo.

**Table 2 nutrients-17-02455-t002:** Differences in mean frequency (MF), median frequency (MDF), and root mean square (RMS) between caffeine and placebo conditions during muscular power tests in the bench press and back squat exercises.

Indicator		25% 1RM			50% 1RM			75% 1RM			90% 1RM			*p* Value (*pη*^2^)		
		Caffeine	Placebo	*p* Value	Caffeine	Placebo	*p* Value	Caffeine	Placebo	*p* Value	Caffeine	Placebo	*p* Value	Condition	Load	Interaction
MF (Hz)	**bench press**															
	*pectoralis major*	127.38 ± 15.53	122.05 ± 12.80	0.137	129.60 ± 12.42	123.17 ± 11.96	0.254	126.11 ± 11.90	120.32 ± 10.39	0.407	199.96 ± 8.70	111.37 ± 16.18	0.197	0.094 (0.18)	**<0.05 (0.32)**	0.955 (0.014)
	*anterior deltoid*	105.37 ± 11.83	104.92 ± 10.22	0.892	111.70 ± 12.82	109.83 ± 13.31	0.582	112.56 ± 14.17	107.75 ± 12.71	0.154	107.70 ± 12.45	105.29 ± 10.73	0.432	0.286 (0.08)	**<0.05 (0.35)**	0.229 (0.086)
	*posterior deltoid*	104.17 ± 15.65	110.38 ± 11.80	0.123	110.04 ± 18.42	114.78 ± 15.64	0.472	103.73 ± 16.68	109.61 ± 14.38	0.331	105.11 ± 10.84	108.51 ± 18.71	0.472	0.147 (0.14)	0.366 (0.07)	0.643 (0.043)
	*biceps brachii*	109.77 ± 17.51	100.85 ± 9.29	**<0.05**	112.23 ± 15.69	102.64 ± 15.65	**<0.05**	112.66 ± 21.66	107.62 ± 17.26	0.193	112.66 ± 15.78	106.16 ± 19.64	0.066	**<0.05 (0.39)**	0.367 (0.07)	0.684 (0.032)
	*lateral head of the triceps*	120.91 ± 10.94	122.92 ± 14.62	0.309	120.21 ± 13.95	127.45 ± 19.62	0.231	120.37 ± 14.08	124.95 ± 17.04	0.504	117.46 ± 14.48	121.60 ± 19.99	0.349	0.339 (0.06)	**<0.05 (0.24)**	0.203 (0.106)
	**back squat**															
	*gluteus maximus*	92.97 ± 12.17	94.63 ± 15.28	0.668	94.57 ± 12.87	96.07 ± 14.15	0.713	98.49 ± 15.04	96.60 ± 13.83	0.674	96.30 ± 16.72	96.19 ± 19.44	0.987	0.960 (0.00)	0.220 (0.09)	0.914 (0.019)
	*rectus femoris*	140.56 ± 7.76	142.41 ± 12.79	0.554	142.03 ± 7.76	142.51 ± 15.0	0.887	141.49 ± 8.84	143.40 ± 10.81	0.37	133.20 ± 21.15	135.40 ± 12.60	0.625	0.514 (0.03)	**<0.05 (0.33)**	0.997 (0.004)
	*vastus medialis*	138.26 ± 12.37	126.71 ± 12.81	**<0.05**	138.99 ± 11.88	133.48 ± 14.93	**<0.05**	138.18 ± 10.94	130.51 ± 14.16	**<0.05**	133.26 ± 22.47	129.68 ± 16.59	0.575	**<0.05 (0.27)**	0.098 (0.11)	0.238 (0.085)
	*vastus lateralis*	114.02 ± 16.02	112.71 ± 15.59	0.762	119.17 ± 17.34	114.18 ± 15.41	0.284	118.78 ± 17.72	115.10 ± 15.25	0.488	114.84 ± 19.35	111.84 ± 15.27	0.553	0.419 (0.04)	**<0.05 (0.17)**	0.485 (0.033)
	*biceps femoris*	124.31 ± 14.85	123.43 ± 13.54	0.869	135.62 ± 20.21	126.86 ± 15.59	0.133	140.70 ± 22.91	132.27 ± 17.21	0.151	130.12 ± 27.96	129.07 ± 17.45	0.906	0.241 (0.09)	**<0.05 (0.20)**	0.726 (0.036)
	*tibialis anterior*	180.26 ± 21.10	175.47 ± 15.00	0.395	174.91 ± 18.87	173.03 ± 20.20	0.755	177.43 ± 20.48	174.36 ± 19.35	0.572	168.59 ± 35.00	174.98 ± 18.96	0.321	0.720 (0.01)	0.588 (0.05)	0.413 (0.064)
MDF (Hz)	**bench press**															
	*pectoralis major*	96.26 ± 19.67	90.01 ± 13.74	0.187	97.90 ± 14.52	85.84 ± 13.20	**<0.05**	95.16 ± 12.93	85.22 ± 11.19	**<0.05**	89.03 ± 8.56	77.46 ± 18.50	**<0.05**	**<0.05 (0.33)**	**<0.05 (0.33)**	0.690 (0.03)
	*anterior deltoid*	21.80 ± 11.06	78.37 ± 6.18	0.264	84.84 ± 13.88	82.62 ± 11.10	0.622	85.95 ± 14.28	80.55 ± 8.13	0.113	81.34 ± 12.74	79.60 ± 8.77	0.496	0.187 (0.11)	**<0.05 (0.22)**	0.206 (0.09)
	*posterior deltoid*	62.51 ± 14.45	66.60 ± 9.21	0.855	71.23 ± 13.40	74.35 ± 11.75	0.966	65.62 ± 13.26	73.93 ± 16.91	0.799	68.75 ± 11.84	74.47 ± 16.41	0.933	0.115 (0.17)	**<0.05 (0.17)**	0.306 (0.08)
	*biceps brachii*	76.91 ± 17.44	67.71 ± 8.68	**<0.05**	80.39 ± 15.59	72.21 ± 14.90	**<0.05**	79.66 ± 22.15	76.34 ± 15.56	**<0.05**	80.52 ± 16.21	76.52 ± 20.49	0.293	**<0.05 (0.29)**	0.105 (0.13)	0.414 (0.06)
	*lateral head of the triceps*	89.38 ± 10.37	93.16 ± 14.23	0.295	90.99 ± 11.95	99.25 ± 19.39	0.120	90.77 ± 13.16	94.94 ± 16.19	0.317	90.18 ± 11.58	93.78 ± 17.99	0.378	0.232 (0.09)	**<0.05 (0.23)**	0.350 (0.07)
	**back squat**															
	*gluteus maximus*	59.18 ± 10.69	59.80 ± 12.01	0.790	62.14 ± 11.72	61.53 ± 11.59	0.830	66.36 ± 14.19	63.70 ± 12.87	0.470	67.55 ± 13.48	66.79 ± 13.28	0.840	0.810 (0.01)	**<0.05 (0.42)**	0.850 (0.03)
	*rectus femoris*	105.40 ± 9.76	108.20 ± 14.09	0.235	108.40 ± 10.10	108.86 ± 16.20	0.885	109.57 ± 11.71	110.72 ± 15.20	0.635	103.24 ± 19.27	105.03 ± 16.57	0.623	0.470 (0.04)	**<0.05 (0.20)**	0.984 (0.01)
	*vastus medialis*	107.22 ± 14.25	96.56 ± 12.77	**<0.05**	109.02 ± 15.08	103.90 ± 15.94	0.099	108.86 ± 12.59	102.54 ± 14.12	0.060	104.86 ± 20.68	104.40 ± 14.46	0.918	**<0.05 (0.24)**	0.163 (0.11)	0.065 (0.16)
	*vastus lateralis*	88.59 ± 13.00	85.34 ± 13.34	0.333	94.15 ± 14.37	88.23 ± 15.53	0.209	94.49 ± 15.80	89.14 ± 14.10	0.238	90.96 ± 17.44	88.47 ± 13.18	0.625	0.267 (0.08)	**<0.05 (0.23)**	0.509 (0.03)
	*biceps femoris*	84.35 ± 13.28	83.38 ± 13.90	0.808	97.98 ± 20.74	86.31 ± 17.90	0.080	105.36 ± 21.76	92.72 ± 19.53	0.059	95.32 ± 23.53	95.17 ± 17.43	0.984	0.129 (0.15)	**<0.05 (0.30)**	0.292 (0.08)
	*tibialis anterior*	155.47 ± 29.34	148.66 ± 21.81	0.408	148.81 ± 28.73	146.98 ± 26.89	0.923	151.40 ± 27.26	150.51 ± 25.90	0.895	145.51 ± 37.48	151.50 ± 25.64	0.311	0.841 (0.00)	0.839 (0.03)	0.467 (0.06)
RMS	**bench press**															
	*pectoralis major*	0.500 ± 0.172	0.428 ± 0.115	**<0.05**	0.495 ± 0.124	0.493 ± 0.129	0.961	0.521 ± 0.156	0.588 ± 0.368	0.541	0.539 ± 0.113	0.543 ± 0.134	0.971	0.695 (0.011)	**<0.05 (0.22)**	0.173 (0.10)
	*anterior deltoid*	0.509 ± 0.116	0.422 ± 0.186	**<0.05**	0.535 ± 0.116	0.482 ± 0.157	0.163	0.541 ± 0.094	0.512 ± 0.144	0.345	0.586 ± 0.090	0.534 ± 0.150	0.124	0.073 (0.199)	0.46 (0.06)	0.408 (0.06)
	*posterior deltoid*	0.514 ± 0.124	0.480 ± 0.124	0.272	0.595 ± 0.155	0.530 ± 0.116	0.097	0.646 ± 0.165	0.605 ± 0.103	0.327	0.686 ± 0.186	0.633 ± 0.136	0.372	0.184 (0.114)	**<0.05 (0.66)**	0.852 (0.03)
	*biceps brachii*	0.301 ± 0.139	0.251 ± 0.132	0.156	0.363 ± 0.173	0.316 ± 0.161	0.387	0.373 ± 0.163	0.338 ± 0.214	0.564	0.371 ± 0.209	0.348 ± 0.229	0.720	0.366 (0.055)	**<0.05 (0.25)**	0.964 (0.01)
	*lateral head of the triceps*	0.325 ± 0.115	0.263 ± 0.078	0.071	0.427 ± 0.128	0.367 ± 0.076	0.110	0.463 ± 0.103	0.432 ± 0.100	0.434	0.493 ± 0.090	0.509 ± 0.121	0.695	0.244 (0.089)	**<0.05 (0.47)**	**<0.05 (0.17)**
	**back squat**															
	*gluteus maximus*	0.308 ± 0.131	0.256 ± 0.100	**<0.05**	0.442 ± 0.154	0.365 ± 0.134	**<0.05**	0.539 ± 0.169	0.446 ± 0.111	**<0.05**	0.567 ± 0.154	0.517 ± 0.154	0.153	**< 0.05 (0.29)**	**<0.05 (0.80)**	0.450 (0.056)
	*rectus femoris*	0.442 ± 0.116	0.376 ± 0.134	0.082	0.448 ± 0.110	0.420 ± 0.147	0.448	0.521 ± 0.134	0.501 ± 0.158	0.691	0.590 ± 0.200	0.526 ± 0.113	0.187	0.100 (0.17)	**<0.05 (0.44)**	0.527 (0.05)
	*vastus medialis*	0.514 ± 0.124	0.480 ± 0.124	0.272	0.595 ± 0.155	0.530 ± 0.116	0.097	0.646 ± 0.165	0.605 ± 0.103	0.327	0.686 ± 0.186	0.633 ± 0.136	0.372	0.184 (0.11)	**<0.05 (0.66)**	0.852 (0.03)
	*vastus lateralis*	0.670 ± 0.737	0.465 ± 0.214	0.184	0.590 ± 0.142	0.548 ± 0.189	0.292	0.640 ± 0.138	0.612 ± 0.216	0.313	0.682 ± 0.182	0.646 ± 0.203	0.315	0.419 (0.04)	**<0.05 (0.66)**	0.467 (0.05)
	*biceps femoris*	0.359 ± 0.264	0.269 ± 0.123	0.217	0.504 ± 0.344	0.315 ± 0.095	**0.033**	0.527 ± 0.338	0.605 ± 0.103	0.063	0.488 ± 0.297	0.633 ± 0.136	0.861	0.820 (0.01)	**<0.05 (0.87)**	**0.003 (0.65)**
	*tibialis anterior*	0.428 ± 0.529	0.257 ± 0.174	0.195	0.301 ± 0.227	0.232 ± 0.111	0.204	0.324 ± 0.134	0.314 ± 0.184	0.804	0.314 ± 0.132	0.321 ± 0.186	0.866	0.172 (0.12)	0.257 (0.08)	0.212 (0.09)

MDF: median frequency, MF: mean frequency, RMS: root mean square. Bold text represents statistically significant results.

**Table 3 nutrients-17-02455-t003:** Differences in ratings of perceived exertion (RPE) and pain perception (PP) between caffeine and placebo conditions during maximal strength tests in the bench press and back squat.

Indicators		Caffeine	Placebo
RPE (6–20)	**bench press**	15 ± 3	16 ± 3
	**back squat**	17 ± 3	16 ± 3
PP (0–10)	**bench press**	1 ± 2	1 ± 1
	**back squat**	1 ± 2	1 ± 2

RPE: ratings of perceived exertion, PP: pian perception.

**Table 4 nutrients-17-02455-t004:** Differences in ratings of perceived exertion (RPE) and pain perception (PP) between caffeine and placebo conditions during maximal strength tests in the bench press and back squat exercises.

Indicator		25% 1RM			50% 1RM			75% 1RM			90% 1RM			*p* Value (*pη*^2^)		
		Caffeine	Placebo	*p* Value	Caffeine	Placebo	*p* Value	Caffeine	Placebo	*p* Value	Caffeine	Placebo	*p* Value	Condition	Load	Interaction
RPE (6–20)	**bench press**	7 ± 1	7 ± 1	0.261	10 ± 2	9 ± 2	0.757	12 ± 2	12 ± 2	0.406	14 ± 3	14 ± 3	0.652	0.497 (0.03)	**<0.05 (0.84)**	0.893 (0.02)
	**back squat**	7 ± 1	7 ± 1	0.544	10 ± 2	10 ± 2	0.648	11 ± 2	12 ± 2	0.205	13 ± 2	14 ± 3	0.076	0.331 (0.06)	**<0.05 (0.86)**	0.142 (0.10)
PP (0–10)	**bench press**	0 ± 1	0 ± 1	0.237	1 ± 1	0 ± 1	0.774	1 ± 1	1 ± 1	0.300	1 ± 1	1 ± 1	0.333	0.158 (0.13)	**<0.05 (0.22)**	0.921 (0.01)
	**back squat**	0 ± 1	1 ± 1	0.204	1 ± 1	1 ± 1	0.158	1 ± 1	1 ± 1	0.325	1 ± 1	1 ± 1	0.105	0.155 (0.13)	**<0.05 (0.21)**	0.659 (0.04)

RPE: ratings of perceived exertion, PP: pian perception. Bold text represents statistically significant results.

**Table 5 nutrients-17-02455-t005:** The number (frequency) of participants who reported side effects immediately after and 24 h later, out of 16 participants.

Side Effect	Caffeine		Placebo	
	+0	+24	+0	+24
Muscle soreness	6 (37.5%)	8 (50%)	6 (37.5%)	7 (43.75%)
Increased urine output	6 (37.5%)	6 (37.5%) *	2 (12.5%)	0 (0%)
Tachycardia and heart palpitations	5 (31.25%)	0 (0%)	1 (6.25)	1 (6.25%)
Anxiety or nervousness	2 (12.5%)	1 (6.25%)	1 (6.25%)	0 (0%)
Headache	0 (0%)	0 (0%)	0 (0%)	0 (0%)
Gastrointestinal problems	2 (12.5%)	3 (18.75%)	0 (0%)	0 (0%)
Insomnia	-	7 (43.75%)	-	2 (12.5%)
Increased vigor/activeness	14 (87.5%) *	5 (31.25%)	4 (25%)	4 (25%)
Perception of performance improvement	14 (87.5%) *	-	2 (12.5%)	-

* *p* < 0.05 vs. placebo.

## Data Availability

Data are available from the corresponding author upon reasonable request.

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
