# Peer review of "Effect of Caffeinated Chewing Gum on Maximal Strength, Muscular Power, and Muscle Recruitment During Bench Press and Back Squat Exercises"

_nutrients, 2025, doi:10.3390/nu17152455_

Round 1
Reviewer 1 Report
Comments and Suggestions for Authors
I thank the authors for an interesting and practically valuable study. I would like to offer a few comments after reading the manuscript.
Introduction: I suggest highlighting the practical relevance of this study in relation to its objective.
Materials and Methods: The authors use figures with one, two, or three decimal places. I recommend standardizing the presentation of numbers to three decimal places.
The date of issue of the Scientific Research Ethics Committee of Shanghai University of Sport (No. 102772023RT203) should be provided.
The study involves a small sample size—16 subjects in 2 groups of 8. For studies of this type, each group should ideally include 12-16 subjects. It is noted that “Low-powered studies (e.g., <20 subjects per group) are prone to overestimating effect sizes and yielding non-replicable findings.” https://doi.org/10.1038/nrn3475. I suggest reclassifying this study as a pilot.
The methodology description states that participants completed the Side Effects Questionnaire, a nine-item dichotomous scale assessing potential caffeine side effects [49], immediately and 24 hours after exercise sessions (2016 line). You have not presented or analyzed these data; please include and discuss them.
Statistical Analyses: You list many methods used. You mention applying McNemar's test (235 line), but the data are not presented and not analyzed. Please provide this information.
Results: The analysis includes many indicators, but they are not explained (e.g., lines 270-277). I suggest commenting on them. Table 2 shows p-values, but the methodology specifies a confidence level of p<0.05. I recommend aligning the presentation with the methodology. This also applies to the discussion of this criterion in the text.
Discussion: I suggest discussing the practical value of the study.
PS: It is necessary to check the grammar of the text, as proofreading errors can occur.
Sincerely.
Author Response
COMMENT #1:
Reviewer #1: Introduction: I suggest highlighting the practical relevance of this study in relation to its objective.
RESPONSE #1
We thank the reviewer for raising this comment. We have included practical relevance in the introduction (Lines 84-89), as follows: “This study aims to investigate the effects of caffeinated chewing gum on maximal strength, muscular power, and neural drive to the prime movers during bench press and back squat in resistance-trained men. We hypothesize that caffeinated chewing gum will improve maximal strength and muscular power by increasing muscle recruitment in both exercises. Our research provides sEMG-based insights into the neuromuscular mechanisms underlying the effects of caffeine on resistance exercise and offers practical guidance to optimize caffeine supplementation in resistance training.”
COMMENT #2:
Materials and Methods: The authors use figures with one, two, or three decimal places. I recommend standardizing the presentation of numbers to three decimal places. .
RESPONSE #2:
We thank Reviewer 1 for this comment. In line with reporting conventions, ratings of perceived exertion and pain perception should be reported as whole numbers, while age, weight, height, and percentages should use one decimal place (Ding et al., 2025; Pallares et al., 2013; Lei et al., 2023). For effect sizes, MF and MDF should be reported with two decimal places, while RMS and F-values should be reported with three decimal places (Grgic et al., 2020; Amir et al., 2017). Additionally, in response to Comment 7, p-values are retained to three decimal places unless they are less than 0.05. The manuscript has been revised accordingly.
Lei TH,Qin Q,Girard O, et al. Caffeine intake enhances peak oxygen uptake and performance during high-intensity cycling exercise in moderate hypoxia. Eur J Appl Physiol. 2024;124 (2):537-549. doi:10.1007/s00421-023-05295-0
Ding L,Liu J,Yao Y, et al. Caffeinated chewing gum enhances maximal strength and muscular endurance during bench press and back squat exercises in resistance-trained men. Front Nutr. 2025;12:1540552. doi:10.3389/fnut.2025.1540552
Pallarés, J.G,Fernández-Elías, V.E,Ortega, J.F,Muñoz, G.,Muñoz-Guerra, J., & Mora-Rodríguez, R. (2013). Neuromuscular responses to incremental caffeine doses: performance and side effects. Medicine and science in sports and exercise, 45 (11), 2184-92. https://doi.org/10.1249/MSS.0b013e31829a6672
Grgic J,Sabol F,Venier S, et al. What Dose of Caffeine to Use: Acute Effects of 3 Doses of Caffeine on Muscle Endurance and Strength. Int J Sports Physiol Perform. 2020;15 (4):470-477. doi:10.1123/ijspp.2019-0433
Pourmoghaddam A,Dettmer M,Malanka SJK, et al. Assessing multiple muscle activation during squat movements with different loading conditions - an EMG study. Biomed Tech (Berl). 2018;63 (4):413-420. doi:10.1515/bmt-2016-0226
COMMENT #3:
The date of issue of the Scientific Research Ethics Committee of Shanghai University of Sport (No. 102772023RT203) should be provided.
RESPONSE #3:
We sincerely appreciate the comment. We have added the dates as follows (Lines 111) “The experimental procedures were approved by the Scientific Research Ethics Committee of Shanghai University of Sport (No. 102772023RT203) on March 15, 2024.”
COMMENT #4:
The study involves a small sample size—16 subjects in 2 groups of 8. For studies of this type, each group should ideally include 12-16 subjects. It is noted that “Low-powered studies (e.g., <20 subjects per group) are prone to overestimating effect sizes and yielding non-replicable findings.” https://doi.org/10.1038/nrn3475. I suggest reclassifying this study as a pilot.
RESPONSE #4:
We thank this reviewer for this comments. We clarify that our study employed a randomized crossover experimental design, with both the placebo and caffeine conditions including 16 participants. A prior power analysis indicated that a minimum of 10 participants was sufficient, making our final sample adequate. Additionally, previous studies with similar research designs have used comparable sample sizes ranging from 12 to 20 (Goldstein et al., 2010; Grgic & Mikulic, 2017; Venier et al., 2019a). The sample size estimation can be found in the methods section of the manuscript, as follows (Line 93-96): “ A priori power analysis using G*Power software (version 3.1; Universität Düsseldorf, Düsseldorf, Germany), indicated that 10 participants would be sufficient (alpha: 0.05, power: 0.80, correlation coefficient: 0.90 [26], and effect size: 0.18-1.28 for 1RM and muscular power in resistance exercise [14,26,31]).”
Venier, S., Grgic, J., & Mikulic, P. (2019a). Acute Enhancement of Jump Performance, Muscle Strength, and Power in Resistance-Trained Men After Consumption of Caffeinated Chewing Gum. Int J Sports Physiol Perform, 14(10), 1415-1421. https://doi.org/10.1123/ijspp.2019-0098
Goldstein, E., Jacobs, P. L., Whitehurst, M., Penhollow, T., & Antonio, J. (2010). Caffeine enhances upper body strength in resistance-trained women. J Int Soc Sports Nutr, 7, 18. https://doi.org/10.1186/1550-2783-7-18
Grgic, J., & Mikulic, P. (2017). Caffeine ingestion acutely enhances muscular strength and power but not muscular endurance in resistance-trained men. Eur J Sport Sci, 17(8), 1029-1036. https://doi.org/10.1080/17461391.2017.1330362
COMMENT #5:
The methodology description states that participants completed the Side Effects Questionnaire, a nine-item dichotomous scale assessing potential caffeine side effects [49], immediately and 24 hours after exercise sessions (2016 line). You have not presented or analyzed these data; please include and discuss them. Statistical Analyses: You list many methods used. You mention applying McNemar's test (235 line), but the data are not presented and not analyzed. Please provide this information.
RESPONSE #5:
We thank Reviewer 1 for this comment. Our study employed McNemar's test to analyze differences in side effects between the placebo and caffeine groups. These results are presented in Table 5, and were previously presented in the manuscript, as follows (Line 368-373): “Caffeinated chewing gum significantly increased perceptions of performance improvement and feelings of vigor/activeness compared to placebo (all p < 0.05), but did not increase other side effects (e.g., headache, muscle soreness, and anxiety) (all p > 0.05) (Table 5). At 24-h post-exercise, it increased urine output (p < 0.05), with no significant effects on other side effects (e.g., headache, muscle soreness, and insomnia) (all p > 0.05) (Table 5)”.
COMMENT #6:
Results: The analysis includes many indicators, but they are not explained (e.g., lines 270-277). I suggest commenting on them.
RESPONSE #6:
We thank Reviewer 1 for this comment. The indicators mentioned in Lines 270-277 – mean power output, peak power output, mean velocity, and peak velocity –are commonly used to assess muscular power in resistance training, as supported by prior studies (Venier et al., 2019; Pallares et al., 2013). These indicators are clearly defined in the introduction (Lines 56-61) and methods section (Lines 169-173) of the manuscript, and are further discussed in detail in the discussion section (Lines 420-432), as follows:
Line 56-61: “Recent evidence indicates that caffeine (3-6 mg/kg) significantly enhances maximal strength (one-repetition maximum [1RM]) [5,10,13] and muscular power (barbell velocity and/or power output) at low-to-high loads (25-75% 1RM) [14,15] in the bench press and back squat exercises for resistance-trained individuals, with more pronounced effects typically observed in the back squat [10,15].”
Line 170-174: “The GymAware Power Testing system (Kinetic Performance Technologies, Canberra, Australia), which has demonstrated excellent reliability in measuring velocity and power output [41], was used to record bar displacement during the concentric phase, including mean velocity (MV in m/s), peak velocity (PV in m/s), mean power output (MPO in W), and peak power output (PPO in W).”
Line 430-442: “Our study showed that 4 mg/kg of caffeine in gum significantly improved MV at 25-90% 1RM during back squat (Figure 3). In contrast, a previous study found that caffeine (3 mg/kg) in capsule form significantly increased MV at 25-75% 1RM during back squat, with higher doses (6-9 mg/kg) required to enhance MV at 90% 1RM (10). This suggests that caffeinated chewing gum may enhance muscular power at higher loads (90% 1RM) with a lower dose (4 vs. 6-9 mg/kg) during back squat compared to capsules [14]. Additionally, our results showed that caffeinated chewing gum significantly increased PV at 25-75% 1RM in the bench press and 25-90% 1RM in the back squat in resistance-trained men (Figure 3). This contrasts with Ruiz-Fernández et al [15], who found no PV improvements across similar intensities. This discrepancy may be attributed to the inclusion of female participants (10 out of 20) in their study, while we exclusively recruited males. Hormonal fluctuations during the menstrual cycle may impair caffeine metabolism, potentially reducing its ergogenic effects in women [59].”
Venier, S., Grgic, J., & Mikulic, P. (2019a). Acute Enhancement of Jump Performance, Muscle Strength, and Power in Resistance-Trained Men After Consumption of Caffeinated Chewing Gum. Int J Sports Physiol Perform, 14(10), 1415-1421. https://doi.org/10.1123/ijspp.2019-0098
Pallarés, J.G,Fernández-Elías, V.E,Ortega, J.F,Muñoz, G.,Muñoz-Guerra, J., & Mora-Rodríguez, R. (2013). Neuromuscular responses to incremental caffeine doses: performance and side effects. Medicine and science in sports and exercise, 45 (11), 2184-92. https://doi.org/10.1249/MSS.0b013e31829a6672
COMMENT #7:
Table 2 shows p-values, but the methodology specifies a confidence level of p<0.05. I recommend aligning the presentation with the methodology. This also applies to the discussion of this criterion in the text.
RESPONSE #7:
We thank Reviewer 1 for their valuable suggestions. We have revised the description of p-values in the manuscript. Specifically, we will retain three decimal places for p-values > 0.05; otherwise, we will indicate them as p < 0.05.
COMMENT #8:
I suggest discussing the practical value of the study.
RESPONSE #8:
We fully agree with Reviewer 1's suggestion. We have added a Practical Application section, as follows (Lines 509-514): “Our findings demonstrate that chewing caffeinated gum (4 mg/kg) for 5 min prior to training significantly increases 1RM, muscular power and muscle recruitment during resistance exercises. Furthermore, given its time efficiency and minimal side effects, caffeinated gum is recommended as an effective and safe supplementation strategy to optimize resistance training performance.”

Reviewer 2 Report
Comments and Suggestions for Authors
Dear authors,
I have reviewed an interesting paper. As a reviewer, I am sending you below my observations, recommendations, and misunderstandings for which I would like you to provide me with arguments. Thank you in advance
AIM
L23-25
This research aims to examine the effect of caffeinated chewing gum on maximal strength, muscular power, and neural drive to prime movers during bench press and back squat exercises
L82-84
This study aims to investigate the effects of caffeinated chewing gum on maximal strength, muscular power, and neural drive to the prime movers during bench press and back squat in resistance-trained men.
Please maintain the same statement of purpose throughout the manuscript
ABSTRACT
- I do not recommend abbreviations in the abstract. However, if they are used, they should be explained when they first appear.
INTRODUCTION
To support this section, I also recommend the following studies, to highlight the complexity of caffeine use in different situations, in different categories of people, so as to justify the inclusion of this supplement in performing specific sports activities
https://doi.org/10.3390/nu16091253,
https://www.mdpi.com/2076-3417/12/6/3197
MATERIALS and METHODS
- Were 5 minutes of chewing enough to assimilate the caffeine and have such rapid effects?
- Simple size in 16 subjects ....this aspect should be well argued! I noticed that the authors refer to 7 participants, according to certain authors, but it should be argued more strongly, taking into account that the idea of ​​this article is different from that of Javier Raya et al. [28]).
L97: I don't understand this aspect: "(41.4 ± 45.1 mg/day"??? ± 45 mg / day deviation from 41.4???
L100-103: the inclusion criteria are presented. Were there also exclusion criteria? If yes, what were those?
L112: "received either caffeinated gum (4 mg/kg)"!!! what kind of chewing gum did you use as long as the subjects are around 70Kg, and the authors claim that they applied 4 mg/kg? How did they dose the gum according to the subjects' weight? Were there gums with different concentrations of caffeine? At 70 kg x 4 mg/kg the authors had chewing gums with 280mg of caffeine in them?????? Or I don't understand the format in which the design is presented!!! I noticed that the authors presented in L202-210, but the assimilation of caffeine in 5 minutes? It is possible that in 5 minutes it is assimilated directly into the blood and then expresses its effects, knowing that caffeine is metabolized by the liver! I ask for clear reasoning, to ensure the reproducibility and scientific validation of the design
L151: what was "pain perception" measured with?? I would like a clarification of L147-154, because if someone wanted to reproduce this part, they could not because it is not clear, detailed
RESULTS
Figures 3 and 4 should have a higher quality, being difficult to analyze and interest
Table 2 should be divided into two, the functional values ​​in one table, the perceived values ​​(RPE and PP) in another table, being different aspects of the subjects' assessment. I recommend. The authors will decide.
Discussion
L352-353 – does not represent a discussion, in this written form, but rather a conclusion. It should be reformulated
I would like the results to be be more highlighted in relation to other recently published studies. The authors have in this section sufficient data from the Results section to discuss in comparison with other studies
Limitation
L471-478 – the information included here is not a limitation. It is not related to the title of this study. It is only assumptions or possible interpretations. Please reformulate and focus on the limitations.
CONCLUSION
ok
REFERENCES
1. What is the difference between the present study (Effect of Caffeinated Chewing Gum on Maximal Strength, Muscular Power and Muscle Recruitment during Bench Press and Back Squat Exercises) and reference no. 23? (Ding, L.; Liu, J.; Yao, Y.; Guo, L.; Chen, B.; Cao, Y.; Girard, O. Caffeinated chewing gum enhances maximal strength and muscular endurance during bench press and back squat exercises in resistance-trained men. Frontiers in Nutrition 2025.)
2. 35% of the references are older than 10 years, some from the 70s and 80s. I'm not saying it's good or bad, I'll let the authors decide
Author Response
COMMENT #1:
AIM
L23-25
This research aims to examine the effect of caffeinated chewing gum on maximal strength, muscular power, and neural drive to prime movers during bench press and back squat exercises
L82-84
This study aims to investigate the effects of caffeinated chewing gum on maximal strength, muscular power, and neural drive to the prime movers during bench press and back squat in resistance-trained men.
Please maintain the same statement of purpose throughout the manuscript
RESPONSE #1
Thank you to Reviewer 2 for the suggestion. We have unified the wording, as follows (Line 23-25): “Abstract: Background/Objectives: This study aims to investigate the effects of caffeinated chewing gum on maximal strength, muscular power, and neural drive to the prime movers during bench press and back squat in resistance-trained men.”
COMMENT #2:
ABSTRACT
- I do not recommend abbreviations in the abstract. However, if they are used, they should be explained when they first appear.
RESPONSE #2:
Thank you to Reviewer 2 for the suggestion. We have added the full names of all abbreviations upon first use, as follows (Line 28-31): “After chewing for 5 min, participants performed a maximal strength test followed by muscular power assessments at 25%, 50%, 75%, and 90% of their one-repetition maximum (1RM), completing with 3, 2, 1, and 1 repetition (s), respectively, for bench press and back squat. Surface electromyography sEMG data were recorded for each repetition.”
COMMENT #3:
INTRODUCTION
To support this section, I also recommend the following studies, to highlight the complexity of caffeine use in different situations, in different categories of people, so as to justify the inclusion of this supplement in performing specific sports activities
https://doi.org/10.3390/nu16091253,
https://www.mdpi.com/2076-3417/12/6/3197
RESPONSE #3:
Thank you for the reviewer’s suggestion. We have added the relevant information in the introduction section as follows:
Line 48-50: “The bench press and back squat are commonly prescribed in strength and conditioning programs to develop upper and lower body muscle strength and power [1,2].”
Line 54-57: “ Caffeine (1,3,7-trimethylxanthine), classified as a nutritional supplement by the International Olympic Committee[8], has been shown to enhance aerobic exercise [9], anaerobic exercise [10], and sport-specific performance [11,12] in both males and females.”
COMMENT #4:
MATERIALS and METHODS
Were 5 minutes of chewing enough to assimilate the caffeine and have such rapid effects? I noticed that the authors presented in L202-210, but the assimilation of caffeine in 5 minutes? It is possible that in 5 minutes it is assimilated directly into the blood and then expresses its effects, knowing that caffeine is metabolized by the liver! I ask for clear reasoning, to ensure the reproducibility and scientific validation of the design
RESPONSE #4:
We greatly appreciate the reviewer’s comment. Previous research (Ranchordas et al., 2018; Yildirim et al., 2023; Field et al., 2024) has established a chewing duration of 5 minutes for caffeinated gum, as this period allows approximately 85% of the caffeine to be released and absorbed into the systemic circulation through the oral mucosa (Kamimori et al., 2002). Furthermore, we did not begin testing immediately after chewing the gum. Indeed, a standardized 10-minute warm-up followed, ensuring an appropriate absorption window. Research has confirmed that peak serum caffeine concentrations are reached 10 minutes after chewing (Morris et al., 2019), which is generally considered to be the optimal time for caffeine's effects. Notably, the primary mechanism by which caffeine exerts its effects is through absorption into the bloodstream, where it binds to adenosine receptors in the central and peripheral nervous systems. While the liver metabolizes caffeine, this process is not the reason for caffeine's effects. In fact, its metabolic products can lead to a range of side effects that may counteract the benefits of caffeine.
Ranchordas MK,King G,Russell M, et al. Effects of Caffeinated Gum on a Battery of Soccer-Specific Tests in Trained University-Standard Male Soccer Players. Int J Sport Nutr Exerc Metab. 2018;28 (6):629-634. doi:10.1123/ijsnem.2017-0405
Yildirim UC,Akcay N,Alexe DI, et al. Acute effect of different doses of caffeinated chewing gum on exercise performance in caffeine-habituated male soccer players. Front Nutr. 2023;10:1251740. doi:10.3389/fnut.2023.1251740
Field A,Corr L,Birdsey L, et al. Caffeine Gum Improves Reaction Time but Reduces Composure Versus Placebo During the Extra-Time Period of Simulated Soccer Match-Play in Male Semiprofessional Players. Int J Sport Nutr Exerc Metab. 2024;34 (5):286-297. doi:10.1123/ijsnem.2023-0220
Kamimori GH,Karyekar CS,Otterstetter R, et al. The rate of absorption and relative bioavailability of caffeine administered in chewing gum versus capsules to normal healthy volunteers. Int J Pharm. 2002;234 (1-2):159-67. doi:10.1016/s0378-5173(01)00958-9
Morris C,Viriot SM,Farooq Mirza QUA, et al. Caffeine release and absorption from caffeinated gums. Food Funct. 2019;10 (4):1792-1796. doi:10.1039/c9fo00431a
COMMENT #5:
Simple size in 16 subjects ....this aspect should be well argued! I noticed that the authors refer to 7 participants, according to certain authors, but it should be argued more strongly, taking into account that the idea of ​​this article is different from that of Javier Raya et al. [28]).
RESPONSE #5:
Thank you to this reviewer for the suggestion. To strengthen the persuasiveness of our sample size, we recalculated the required number of participants based on the effect sizes from two other studies with similar experimental designs. These studies have shown that caffeine affects 1RM and muscular power during the bench press and back squat, with effect sizes ranging from 0.18 to 1.28 (Ding et al., 2025; Jesus et al., 2013). Using these values, we performed a sample size calculation using G*Power (α-level: 0.05, 1-β error probability: 0.8, correlation coefficient: 0.9, and effect size (f): 0.18-1.28), which indicated that a sample size of 4 to 10 participants would be sufficient to examine the effect of caffeine on 1RM and muscular power during both exercises. Additionally, many previous studies employing similar research designs have used comparable sample sizes, typically ranging from 12 to 20 (Goldstein et al., 2010; Grgic & Mikulic, 2017; Venier et al., 2019a). In summary, we consider our sample of 16 participants to be adequate.
We have revised wording (Line 93-95), as follows: “A priori power analysis, conducted using G*Power software (version 3.1; Universität Düsseldorf, Düsseldorf, Germany), indicated that 10 participants was sufficient (alpha: 0.05, power: 0.80, correlation coefficient: 0.90 [26], and effect size: 0.18-1.28 for 1RM and muscular power in resistance exercise [14,26,31])”
Tallis, J.,Duncan, M.J,Clarke, N.D,Morris, R.O, & Tamilio, R.A (2024). Are caffeine effects equivalent between different modes of administration: the acute effects of 3 mg.kg -1 caffeine on the muscular strength and power of male university Rugby Union players. Journal of the International Society of Sports Nutrition, 21 (1), 2419385. https://doi.org/10.1080/15502783.2024.2419385
Teimouri-Korani, H., Hemmatinafar, M., Willems, M. E., Rezaei, R., & Imanian, B. (2025). Individual responses to encapsulated caffeine and caffeine chewing gum on strength and power in strength-trained males. Journal of the International Society of Sports Nutrition, 22(1). https://doi.org/10.1080/15502783.2025.2495228
Whalley, P.J,Dearing, C.G, & Paton, C.D (2020). The Effects of Different Forms of Caffeine Supplement on 5-km Running Performance. International journal of sports physiology and performance, 15 (3), 390-394. https://doi.org/10.1123/ijspp.2019-0287
Whalley, P.,Paton, C., & Dearing, C.G (2021). Caffeine metabolites are associated with different forms of caffeine supplementation and with perceived exertion during endurance exercise. Biology of sport, 38 (2), 261-267. https://doi.org/10.5114/biolsport.2020.98455
COMMENT #6:
L97: I don't understand this aspect: "(41.4 ± 45.1 mg/day"??? ± 45 mg / day deviation from 41.4???
RESPONSE #6:
We apologize for the previous calculation error. We recalculated the relevant data, and the result is: 41.4 ± 46.5 mg/day. The parameters for the 16 participants are as follows: 0, 72.5, 0, 36.3, 0, 0, 87.5, 92.5, 72.5, 50, 157.5, 0, 0, 57.3, 0, 36.6 (mg/day). We have corrected this in the original text, as follows (Line 100): “Habitual caffeine intake (41.4 ± 46.5 mg/day) was assessed using a validated self-reported questionnaire [33], and participants were classified as either naive or mild caffeine consumers (0-2.99 mg/kg/day) [34].”
COMMENT #7:
L100-103: the inclusion criteria are presented. Were there also exclusion criteria? If yes, what were those?
RESPONSE #7:
Thank you to the reviewer for the comments. We have added the exclusion criteria, as follows (Line 107-109): “Exclusion criteria included: (a) any neuromuscular, immunological, cardiometabolic, neurological or musculoskeletal disorders; (b) self-reported smoking; or (c) caffeine allergy.”
COMMENT #8:
L112: "received either caffeinated gum (4 mg/kg)"!!! what kind of chewing gum did you use as long as the subjects are around 70Kg, and the authors claim that they applied 4 mg/kg? How did they dose the gum according to the subjects' weight? Were there gums with different concentrations of caffeine? At 70 kg x 4 mg/kg the authors had chewing gums with 280mg of caffeine in them?????? Or I don't understand the format in which the design is presented!!!.
RESPONSE #8:
We apologize for any confusion caused by our previous statements. Consistent with previous research that used caffeinated chewing gum (3 mg/kg) (Shiu et al., 2024; Tasi et al., 2024), we prepared both the caffeinated chewing gum (each piece containing 100 mg of caffeine) and placebo gum by crushing, grinding, blending, and reshaping them. Subsequently, we weighed each gum sample according to the participants' body weight using a high-precision electronic digital scale and wrapped them uniformly in foil. Therefore, this study administered caffeine as a relative dose rather than an absolute dose. We have added these relevant details in the methods section, as follows (Line 212-216): “The caffeine source selected was commercially available Military Energy Gum (Market Right Inc., Plano, IL, USA), containing 100 mg of caffeine per piece. The placebo gum was a caffeine-free version with a similar taste and shape. To achieve a relative dose of 4 mg/kg, which has been shown to enhance performance with minimal side effects [9], all gum samples were crushed, ground, blended, and reshaped. These were then weighed precisely using a high-precision electronic digital scale and wrapped in aluminum foil by a designated experimenter.”
Shiu YJ,Chen CH,Tao WS, et al. Acute ingestion of caffeinated chewing gum reduces fatigue index and improves 400-meter performance in trained sprinters: a double-blind crossover trial. J Int Soc Sports Nutr. 2024;21 (1):2414871. doi:10.1080/15502783.2024.2414871
Tsai MT, Shiu YJ,Ho CC, et al. Effects of Caffeinated Chewing Gum on Ice Hockey Performance after Jet Lag Intervention: Double-Blind Crossover Trial. Nutrients. 2024;16 (18):. doi:10.3390/nu16183151
COMMENT #9:
L151: what was "pain perception" measured with?? I would like a clarification of L147-154, because if someone wanted to reproduce this part, they could not because it is not clear, detailed.
RESPONSE #9:
We apologize for any confusion caused by our unclear statements. We have added details regarding the method for measuring pain perception in the methods section, as follows (Line 155-157): “Participants first performed a warm-up consisting of 10 and 5 repetitions at 50% and 75% 1RM, respectively, as assessed during the familiarization session [5]. Following the recommendations of Baechle and Earle [36], 1RM was determined within 3-5 attempts, with a 5-min rest between successful attempts. Ratings of perceived exertion (RPE) and pain perception were measured by 6-20 Borg scale and a 0-10 numeric pain rating scale [37,38], respectively, within 5 seconds of each successful attempt.”
COMMENT #10:
RESULTS
Figures 3 and 4 should have a higher quality, being difficult to analyze and interest
Table 2 should be divided into two, the functional values ​​in one table, the perceived values ​​(RPE and PP) in another table, being different aspects of the subjects' assessment. I recommend. The authors will decide.
RESPONSE #10:
Thank you to this reviewer for the suggestion; we have recreated higher-resolution images (Figure 3 and 4). Additionally, we have created separate tables for RPE and PP (Table 2 and 4).
COMMENT #11:
Discussion
L352-353 – does not represent a discussion, in this written form, but rather a conclusion. It should be reformulated
I would like the results to be more highlighted in relation to other recently published studies. The authors have in this section sufficient data from the Results section to discuss in comparison with other studies.
RESPONSE #11:
Thank you for the reviewer's comment. However, according to writing conventions (Mumford et al., 2016; Cao et al., 2025; Filip et al., 2012), the first paragraph of the discussion should summarize key results without comparing them to other studies. These comparisons are subsequently presented in the main body of the discussion, as outlined in sections 4.1-4.5 of our manuscript.
Mumford PW, Tribby AC,Poole CN, et al. Effect of Caffeine on Golf Performance and Fatigue during a Competitive Tournament. Med Sci Sports Exerc. 2016;48 (1):132-8. doi:10.1249/MSS.0000000000000753
Cao Y,He W,Ding L, et al. Dose-response effects of caffeine during repeated cycling sprints in normobaric hypoxia to exhaustion. Eur J Appl Physiol. 2025;125 (1):223-236. doi:10.1007/s00421-024-05576-2
Filip-Stachnik A,Krawczyk R,Krzysztofik M, et al. Effects of acute ingestion of caffeinated chewing gum on performance in elite judo athletes. J Int Soc Sports Nutr. 2021;18 (1):49. doi:10.1186/s12970-021-00448-y
COMMENT #12:
Limitation
L471-478 – the information included here is not a limitation. It is not related to the title of this study. It is only assumptions or possible interpretations. Please reformulate and focus on the limitations.
RESPONSE #12:
Thank you for the reviewer's suggestion. We have revised the limitations section, as follows (Line 497-506): “This study has several limitations. First, the results are limited to resistance-trained males, reducing generalizability to other populations such as adolescents, older adults, and females. Second, sEMG provides a global estimate of neuro-muscular activation but cannot distinguish between the number of recruited motor units or their firing frequencies, and crosstalk from neighbouring muscles is possible [66]. Future research should explore motor nerve and/or transcranial magnetic stimulation to assess a broader range of neuro-mechanical parameters, along with varying caffeine ingestion methods and doses across different resistance exercise protocols (i.e., traditional-set vs. cluster-set configurations). Additionally, incorporating transcranial magnetic stimulation-induced motor evoked potentials and muscle oxygenation (near-infrared spectroscopy) could provide insights into cortical and spinal mechanisms of fatigue, as well as metabolic differences (i.e., muscle oxygen delivery) between caffeine ingestion methods and doses [64]. Third, our study did not account for genetic variability in caffeine response. Previous research has found that individuals with the AA genotype are more sensitive to caffeine than those with AC or CC genotypes [9]. Finally, while randomized condition orders, consistent trial times, and dietary instructions were used to minimize variability, the lack of strict control over food intake 2-3 h before trials may have introduced variability between conditions.”
COMMENT #13:
REFERENCES
What is the difference between the present study (Effect of Caffeinated Chewing Gum on Maximal Strength, Muscular Power and Muscle Recruitment during Bench Press and Back Squat Exercises) and reference no. 23? (Ding, L.; Liu, J.; Yao, Y.; Guo, L.; Chen, B.; Cao, Y.; Girard, O. Caffeinated chewing gum enhances maximal strength and muscular endurance during bench press and back squat exercises in resistance-trained men. Frontiers in Nutrition 2025.)
RESPONSE #13:
Thank you for the reviewer's question. The two studies differ mainly in two aspects. First, the outcomes: our previous study focused on the effect of caffeine on muscular endurance (the number of repetitions to failure), whereas this study focuses on muscular power (bar displacement velocity and power output). Secondly, this is the first study to use surface electromyography to assess the impact of caffeine on muscle activity in major muscle groups during the bench press and back squat. This approach clarifies the underlying mechanisms of caffeine's effects and helps explain its greater impact on lower body strength compared to upper body strength. Therefore, we believe that the differences between these two studies are significant.
COMMENT #14:
REFERENCES
35% of the references are older than 10 years, some from the 70s and 80s. I'm not saying it's good or bad, I'll let the authors decide.
RESPONSE #14:
Thank you for the reviewer's suggestion. We have updated several references with more recent literature. That said, some older references are instrumental articles or books that remain essential, so they have been retained.

Round 2
Reviewer 2 Report
Comments and Suggestions for Authors
Dear Authors,
Thank you for the corrections and information provided in response to my recommendations.
I would like these references to remain or be added, along with the existing ones, because they offer strong theoretical and conceptual support.
Ranchordas MK,King G,Russell M, et al. Effects of Caffeinated Gum on a Battery of Soccer-Specific Tests in Trained University-Standard Male Soccer Players. Int J Sport Nutr Exerc Metab. 2018;28 (6):629-634. doi:10.1123/ijsnem.2017-0405
Yildirim UC,Akcay N,Alexe DI, et al. Acute effect of different doses of caffeinated chewing gum on exercise performance in caffeine-habituated male soccer players. Front Nutr. 2023;10:1251740. doi:10.3389/fnut.2023.1251740
Field A,Corr L,Birdsey L, et al. Caffeine Gum Improves Reaction Time but Reduces Composure Versus Placebo During the Extra-Time Period of Simulated Soccer Match-Play in Male Semiprofessional Players. Int J Sport Nutr Exerc Metab. 2024;34 (5):286-297. doi:10.1123/ijsnem.2023-0220
Kamimori GH,Karyekar CS,Otterstetter R, et al. The rate of absorption and relative bioavailability of caffeine administered in chewing gum versus capsules to normal healthy volunteers. Int J Pharm. 2002;234 (1-2):159-67. doi:10.1016/s0378-5173(01)00958-9
Morris C,Viriot SM,Farooq Mirza QUA, et al. Caffeine release and absorption from caffeinated gums. Food Funct. 2019;10 (4):1792-1796. doi:10.1039/c9fo00431a